# Are Signals Regulating Energy Homeostasis Related to Neuropsychological and Clinical Features of Gambling Disorder? A Case–Control Study

**DOI:** 10.3390/nu14235084

**Published:** 2022-11-29

**Authors:** Mikel Etxandi, Isabel Baenas, Bernat Mora-Maltas, Roser Granero, Fernando Fernández-Aranda, Sulay Tovar, Neus Solé-Morata, Ignacio Lucas, Sabela Casado, Mónica Gómez-Peña, Laura Moragas, Amparo del Pino-Gutiérrez, Ester Codina, Eduardo Valenciano-Mendoza, Marc N. Potenza, Carlos Diéguez, Susana Jiménez-Murcia

**Affiliations:** 1Department of Psychiatry, Bellvitge University Hospital-Bellvitge Institute for Biomedical Research (IDIBELL), 08907 Barcelona, Spain; 2Department of Psychiatry, Hospital Universitari Germans Trias i Pujol, IGTP Campus Can Ruti, 08916 Badalona, Spain; 3Ciber Fisiopatología Obesidad y Nutrición (CIBERObn), Instituto de Salud Carlos III, 28029 Madrid, Spain; 4Psychoneurobiology of Eating and Addictive Behaviors Group, Neurosciences Programme, Bellvitge Institute for Biomedical Research (IDIBELL), 08908 Barcelona, Spain; 5Department of Psychobiology and Methodology, Autonomous University of Barcelona, 08193 Barcelona, Spain; 6Department of Clinical Sciences, School of Medicine and Health Sciences, University of Barcelona, 08907 Barcelona, Spain; 7Department of Physiology, CIMUS, Instituto de Investigación Sanitaria, University of Santiago de Compostela, 15782 Santiago de Compostela, Spain; 8Department of Public Health, Mental Health and Perinatal Nursing, School of Nursing, University of Barcelona, 08907 Barcelona, Spain; 9Department of Psychiatry, Yale University School of Medicine, New Haven, CT 06510, USA; 10Child Study Center, Yale University School of Medicine, New Haven, CT 06510, USA; 11Connecticut Mental Health Center, New Haven, CT 06519, USA; 12Connecticut Council on Problem Gambling, Wethersfield, CT 06106, USA; 13Department of Neuroscience, Yale University, New Haven, CT 06520, USA

**Keywords:** gambling disorder, addictive behavior, impulsive–compulsive behavior, gut hormones, adipocytokines, neuropsychology

## Abstract

Gambling disorder (GD) is a modestly prevalent and severe condition for which neurobiology is not yet fully understood. Although alterations in signals involved in energy homeostasis have been studied in substance use disorders, they have yet to be examined in detail in GD. The aims of the present study were to compare different endocrine and neuropsychological factors between individuals with GD and healthy controls (HC) and to explore endocrine interactions with neuropsychological and clinical variables. A case–control design was performed in 297 individuals with GD and 41 individuals without (healthy controls; HCs), assessed through a semi-structured clinical interview and a psychometric battery. For the evaluation of endocrine and anthropometric variables, 38 HCs were added to the 41 HCs initially evaluated. Individuals with GD presented higher fasting plasma ghrelin (*p* < 0.001) and lower LEAP2 and adiponectin concentrations (*p* < 0.001) than HCs, after adjusting for body mass index (BMI). The GD group reported higher cognitive impairment regarding cognitive flexibility and decision-making strategies, a worse psychological state, higher impulsivity levels, and a more dysfunctional personality profile. Despite failing to find significant associations between endocrine factors and either neuropsychological or clinical aspects in the GD group, some impaired cognitive dimensions (i.e., WAIS Vocabulary test and WCST Perseverative errors) and lower LEAP2 concentrations statistically predicted GD presence. The findings from the present study suggest that distinctive neuropsychological and endocrine dysfunctions may operate in individuals with GD and predict GD presence. Further exploration of endophenotypic vulnerability pathways in GD appear warranted, especially with respect to etiological and therapeutic potentials.

## 1. Introduction

Gambling disorder (GD) has been classified as a behavioral addiction (BA) in the Diagnostic and Statistical Manual of Mental Disorders, Fifth Edition (DSM-5) [1], being characterized by recurrent maladaptive gambling behavior, leading to negative consequences in one or more areas of life functioning [2]. Diagnostic criteria include the need to gamble with increasing amounts of money (i.e., tolerance), the tendency to chase losses, irritability when attempting to stop the behavior (i.e., abstinence), the presence of unsuccessful efforts to control gambling behavior, a predominance of thoughts focused on the gambling behavior, the presence of lies or the loss of a significant relationship or job/educational opportunity because of gambling, and the propensity to gamble when feeling distressed or to rely on others to provide money to relieve desperate financial situations caused by gambling [1]. From an etiological perspective, neuroimaging, genetic, and biochemical studies have suggested shared vulnerability factors between addictive-related disorders, such as GD and substance use disorders (SUDs) [3,4]. For instance, dysfunctional neurobiological pathways involved in reward processing [5], which may underlie impulsive and compulsive tendencies, have been described [6].

Several endocrine factors have been implicated in brain responses to rewards and gratification [7,8] including gut hormones (e.g., ghrelin) and adipocytokines (e.g., leptin and adiponectin) [9,10], classically associated with food intake regulation and energy balance [11]. Despite its stimulating appetite role, ghrelin has been described as a hedonic neural reinforcer for natural (e.g., food) and non-natural rewards (e.g., drugs) by its interaction with dopamine signaling in the mesolimbic circuit and other neuroendocrine pathways (e.g., linked to stress, appetite, and metabolic processing) [12]. Ghrelin has been extensively studied in different addictive-related disorders, such as binge eating disorder (BED) and obesity [13,14], as well as in SUDs [15], especially involving alcohol [16,17].

Noticeably, ghrelin up-regulation has been described in SUDs, which positively correlates with craving, abstinence, and relapse [17,18,19]. Accordingly, exogenous ghrelin administration increases craving and drug consumption [9], contrary to ghrelin antagonists [20,21]. An antagonist of ghrelin named liver enriched antimicrobial peptide 2 (LEAP2) has been recently described [22,23]. It has been related to impulsivity and cognitive functioning [24] and may contribute to addictions due to its interplay with ghrelin. Furthermore, genetic alterations related to the ghrelin system, such as receptor polymorphisms, have been associated with reward-seeking behaviors and consumption [25], which together may have potential therapeutic implications [26,27]. In GD, a study by Sztainert et al. [28] suggests ghrelin as a potential predictor of gambling craving and persistence.

Adipocytokines have also been studied in relation to impulsivity [29] and addiction [30,31]. As in the case of regulation of food intake, opposite effects on craving and abstinence have been attributed to leptin compared with ghrelin [32]. Leptin concentrations have been inversely correlated with consumption severity [33], being proposed as a possible biomarker in SUDs involving alcohol and cocaine [10,34]. However, studies regarding alcohol consumption have shown inconsistent results [30], even describing higher leptin concentrations in individuals with alcohol use disorder than in HCs, positively associated with alcohol intake [30,34]. A single study exploring leptin concentrations in GD did not find significant differences compared with those in HCs [35]. Despite there are fewer studies related to adiponectin and addictions, decreased serum concentrations have been reported in obesity with and without eating disorders and in opioid use disorder [14,36]. Adiponectin has also been proposed as a biomarker of craving, like ghrelin, in alcohol use disorder [37]. Similar to other addictive-related disorders, these endocrine substrates represent potential candidates involved in the pathogenesis of BAs, such as GD [35]. However, this area remains underexplored in GD, and further research is needed.

Other neurobiological features linked to addictive-related disorders include impaired neuropsychological processes not only regarding executive functions, such as response inhibition, self-regulation, decision-making, cognitive flexibility, and planning but also working memory [38,39,40,41,42]. These cognitive functions have been described as core symptoms in BAs [39] and are especially related to impulse control [41]. More severe neuropsychological impairment has been described among older patients with GD and preferences for non-strategic gambling [41,43,44]. Neuropsychological impairment has statistically predicted poorer treatment outcome, with more frequent dropout and relapse [41,45].

Beyond neuropsychological factors, other psychological and clinical features have been implicated in the development of addictive disorders [46]. In GD, for example, certain personality traits such as high levels of novelty-seeking (related to impulsivity) and harm avoidance, especially in women [46,47], together with low self-directedness have been linked to both GD and SUDs [48,49]. Difficulties in emotion regulation and poorer psychological states have been linked to GD [50], particularly in women and older individuals with non-strategic gambling [43,44]. A more dysfunctional psychological profile has been associated with greater neuropsychological impairment [41,43,44]. However, studies have largely not explored relationships between different neurobiological features (i.e., endocrine, and neuropsychological factors) and psychological and clinical variables.

To the best of our knowledge, this is the first study that explores the roles of multiple specific signals implicated in addiction and energy homeostasis, meaning food intake and energy expenditure, and clinical and psychological measures among a clinical population with GD. We aimed to explore and compare plasma concentrations of specific metabolic hormones (i.e., leptin, ghrelin, adiponectin, and LEAP2) between patients with GD and HCs. As a second aim, we analyzed correlations between the mentioned endocrine factors and neuropsychological and clinical features. In line with previous literature in addictive disorders, we hypothesized the existence of significant differences in plasma hormonal concentrations between the GD and HC groups. We also hypothesized poorer cognitive functioning, worse psychopathological state, and a more dysfunctional personality profile among individuals with GD. These features were also hypothesized to be related to endocrine alterations, including being able to statistically predict GD presence.

## 2. Materials and Methods

### 2.1. Participants

The sample consisted of *n* = 297 treatment-seeking adult outpatients with GD (93.6% males) with a mean age of 39.58 years (SD = 14.16), voluntarily recruited at the Behavioral Addictions Unit-Psychiatry Department of Bellvitge University Hospital (Barcelona, Spain). As inclusion criteria, all the patients had a diagnosis of GD according to DSM-5 criteria [1]. The HC group was composed by 41 individuals without GD (90.2% males), with a mean age of 49.27 years (SD = 15.23) and was recruited via advertisement from the same catchment area. Regarding anthropometric and endocrine variables, 79 HCs were evaluated by adding to the initial sample 38 healthy adults from CIMUS, University of Santiago de Compostela (Santiago de Compostela, Spain). General exclusion criteria for all participants were the presence of an organic mental disorder, an intellectual disability, a neurodegenerative disorder (such as Parkinson’s disease) or an active psychotic disorder. Recruitment of participants occurred from April 2018 to September 2021, and the evaluation of individuals with GD took place before starting treatment at the Behavioral Addictions Unit-Psychiatry Department of Bellvitge University Hospital (Barcelona, Spain).

Appendix A contains the complete description for the participants in the study.

### 2.2. Measures

#### 2.2.1. Hormonal Assays

Endocrine variables were quantified from peripheral blood sample extraction by venous aspiration with ethylenediamine tetraacetic acid (EDTA; 25 mM final concentration), all samples were collected at 9 am, after at least 8 h of fasting. The blood was centrifuged at 1700 g in a refrigerated centrifuge (4 °C) for 20 min. Plasma was immediately separated from serum and stored at −80 °C until analysis. Parameter determinations were conducted using commercial kits according to the manufacturer’s instructions and in a single analysis to reduce inter-assay variability. The quantitative measurement of LEAP-2 in plasma was performed using a commercial enzyme-linked immunosorbent assay (ELISA) kit (Human LEAP-2 [37,38,39,40,41,42,43,44,45,46,47,48,49,50,51,52,53,54,55,56,57,58,59,60,61,62,63,64,65,66,67,68,69,70,71,72,73,74,75,76] ELISA kit, Phoenix Pharmaceuticals, Inc., Burlingame, CA, USA), previously validated [51,52]. Intra-assay and inter-assay variation coefficients were <10% and <15%, respectively. The assay sensitivity limit was 0.15 ng/mL. Total ghrelin (pg/mL) was measured by ELISA kit (Invitrogen-Thermo Fisher Scientific, Madrid, Spain) for detection of human ghrelin, with a specificity of 100%. Intra-assay variation coefficient was <6% and inter-assay <8.5%. The assay sensitivity limit was 11.8 pg/mL [53]. Adiponectin (ng/mL) and leptin (ng/mL) plasma measurements were performed using a solid-phase sandwich ELISA kit (Invitrogen-Thermo Fisher Scientific, Madrid, Spain) with a specificity of 100%. Intra-assay and inter-assay variation coefficients were <4% and <5%, respectively, and assay sensitivity limit was 100 pg/mL for adiponectin and <3.5 pg/mL for leptin. The absorbance from each sample was measured in duplicate using a spectrophotometric microplate reader at a wavelength of 450 nm (Epoch 2 microplate reader, Biotek Instruments, Inc., Winooski, VT, USA).

#### 2.2.2. Neuropsychological Variables

Iowa Gambling Task (IGT) [54]. A computerized task to evaluate decision-making, risk, reward, and punishment value. The participant must select 100 cards from four decks (i.e., A, B, C, and D), and after each card selection, an output is given either a gain or a loss of money. The participant is instructed that the aim of the task is to win as much money as possible. This test is scored by subtracting the number of cards selected from decks A and B from the number of cards selected from decks C and D. While decks A and B are not advantageous as the final loss is higher than the final gain, decks C and D are advantageous since the punishments are smaller. Higher scores point to better performance, while negative scores point to persistently choosing disadvantageous decks.

Wisconsin Card Sorting Test (WCST) [55] is a task for assessing cognitive flexibility and inhibitory control, composed of four stimulus cards and 128 response cards showing different shapes, colors, and numbers of figures in each one. The participant must match response cards with the stimulus cards in a way that it seems justifiable before receiving the feedback (i.e., correct, or incorrect). After ten sequential correct answers the categorization criteria changes. The number of complete categories, percentage of perseverative errors, and percentage of non-perseverative errors are recorded.

Stroop Color and Word Test (SCWT) [56] consists of three different lists, beginning with a word list containing the names of colors printed in black ink; then, a color list that comprises letter “X” printed in color; and, finally, a color-word list constituted of names of colors in a color ink that does not match the written name. Three final scores are obtained based on the number of items that the participant can read on naming on each list in 45 s. It assesses the ability to inhibit cognitive interference, which occurs when the processing of a stimulus feature affects the simultaneous processing of another attribute of the same stimulus.

Trail Making Test (TMT) [57] consists of 25 circles spread out over two sheets of paper (Parts A and B). The participant is told to connect these circles drawing a line between consecutive numbers (part A) and alternating numbers and letters following a sequential order (part B). The task assesses visual conceptual and visual-motor tracking, entailing motor speed, attention, and the capacity to alternate between cognitive categories (set-shifting). Each part is scored according to the spent time to complete the task.

Digits backward task of the Wechsler Memory Scale-Third Edition (WMS-III) [58] consists of two lists of digits presented verbally by the examiner. The participant is asked to repeat the digits in the same order (first list) and in reverse order (second list). It assesses verbal working memory due to internal manipulation of mnemonic representations of verbal information that is required in the absence of external cues.

Vocabulary subtest of the Wechsler Adult Intelligence Scale, 3rd ed. (WAIS-III) [59] requires defining words of increasing difficulty orally presented, to assess the vocabulary expression and to estimate intellectual capacity [60].

#### 2.2.3. Clinical Variables

South Oaks Gambling Screen (SOGS) [61], Spanish validation [62], is a 20-item instrument for screening past-year gambling problems and related negative consequences. The total score is a measure of problem-gambling severity, with a score of five or more suggestive of “probable pathological gambling”. Its internal consistency in the study sample was Cronbach’s alpha (α) = 0.735.

Diagnostic Questionnaire for Pathological Gambling According to DSM criteria [63], Spanish validation [64], is a self-report questionnaire with 19 items coded in a binary fashion (yes-no), used for diagnosing GD according to the DSM-IV-TR and DSM-5 criteria [1]. Its internal consistency in the study sample was α = 0.796.

Symptom Checklist-90-Revised (SCL-90-R) [65], Spanish validation [66], is a 90-item self-report questionnaire measured on an ordinal 3-point scale, evaluating a broad range of psychological problems and psychopathology, based on nine primary symptomatic dimensions (Somatization, Obsession–Compulsion, Interpersonal Sensitivity, Depression, Anxiety, Hostility, Phobic Anxiety, Paranoid Ideation, and Psychoticism). It includes three global indices (global severity index, positive symptom distress index, and total positive symptom). The internal consistency in the study was α = 0.979.

Temperament and Character Inventory-Revised (TCI-R) [67], Spanish validation [68], is a questionnaire with 240-items scored on a 5-point Likert scale, measuring personality derived from three character dimensions (Self-Directedness, Cooperativeness, and Self-Transcendence) and four temperament dimensions (Harm Avoidance, Novelty Seeking, Reward Dependence, and Persistence). It is used only for research purposes in a public non-profit hospital, in its Spanish adaptation in which the original author participated [68]. The internal consistency in the study was between α = 0.702 (Novelty Seeking) and α = 0.876 (Persistence).

Impulsive Behavior Scale (UPPS-P) [69], Spanish validation [70], measures five facets of impulsive behavior through self-report on 59 items: negative urgency; positive urgency; lack of premeditation; lack of perseverance; and sensation-seeking. The internal consistency in the study was between α = 0.799 (lack of perseverance) and α = 0.928 (positive urgency).

#### 2.2.4. Other Variables

Additional data (e.g., socio-demographic, socio-economic, anthropometric variables, and GD-related characteristics) were collected in a semi-structured face-to-face clinical interview as described elsewhere [71].

### 2.3. Procedure

All patients and HCs from the same catchment area were evaluated at the Behavioral Addictions Unit-Psychiatry Department of Bellvitge University Hospital (Barcelona, Spain), by an expert multidisciplinary team in the field of GD. In the first session, a comprehensive semi-structured clinical interview was conducted, in which all aspects related to gambling behavior were assessed. During the second session, the extraction of blood samples occurred. Samples were analyzed in CIMUS, University of Santiago de Compostela (Santiago de Compostela, Spain), where 38 out of 79 HCs were evaluated regarding endocrine and anthropometric measures. The neuropsychological assessment was performed in a third session.

### 2.4. Statistical Analysis

The statistical analysis was conducted with Stata17 for Windows [72]. Comparison between groups (GD versus HC) were made by Analysis of Covariance (ANCOVA), adjusting for sex, age, and body mass index (BMI) for endocrine variables, and adjusting for sex, age, and education level for neuropsychological variables. The effect size for the mean comparisons was obtained with the standardized Cohen’s-d, considering moderate-mild effect values 0.50 < |d| < 0.80 and high-large effect values |d| > 0.80 [73].

Associations between endocrine and neuropsychological and clinical variables were estimated with partial correlation coefficients, adjusting for sex, age, and BMI (associations with the neuropsychological tasks also included adjustment for the education level). Due to strong associations between the null-significance test for the correlation models given the sample sizes (low correlations achieve significance in large samples, and vice versa), in this study mild-moderate correlation was considered for values |R| > 0.24 and high-large correlation for values |R| > 0.37 [74].

A predictive model was obtained to select the variables with discriminative capacity to identify the presence of GD, through logistic regression. The criterion for the modeling was the diagnosis of GD (presence/absence), and potential predictors included sociodemographic measures, global psychopathological distress, impulsivity levels, personality features, and neuropsychological and endocrine measures. A stepwise selection method was used to automatically select significant contributors. Sex, age, and BMI were included as adjustment/covariables. The goodness-of-fit was measured with the Hosmer–Lemeshow test, the overall predictive capacity with the Cox-Snell’s pseudo-R2, and the overall discriminative capacity with the area under the Receiver Operating Curve (ROC).

In this study, the increase in the Type-I error due to the performance of multiple significance tests was controlled with the familywise error Finner’s procedure, which has shown greater efficiency than the classic Bonferroni adjustment method [75].

## 3. Results

### 3.1. Comparison of Endocrine Measures

Adjusting for sex, age, and BMI, the GD group reported higher ghrelin and lower LEAP2 and adiponectin values compared with HCs (Table 1 and first panel of Figure 1). No differences were found regarding leptin values.

### 3.2. Comparison of Neuropsychological and Clinical Measures

ANCOVAs comparing the mean values of the neuropsychological measures and clinical variables are displayed in Table 2 (see also second panel of Figure 1). Compared to HCs, the GD group displayed worse performance on the WCST, WAIS-vocabulary test, and performance-learning curve during IGT performance (Figure 2). The GD group also reported worse psychopathological states (higher mean scores on the SCL-90 R), higher impulsivity (except on the UPPS-P sensation-seeking scale) and more dysfunctional personality profiles (except on the TCI-R persistence scale).

### 3.3. Associations between Endocrine Variables and Neuropsychological and Clinical Measures

Appendix A displays the partial correlation matrix between the endocrine profile with neuropsychological and clinical variables (psychopathology, impulsivity, and problem-gambling severity). No relevant associations were found within the GD subsample. Among HCs, lower ghrelin values were related to higher values on the IGT-block 5 and, on the WCST scales, number of trials and number of perseverative errors. Higher LEAP2 values were associated with worse psychological states and poorer performance on the TMT, Stroop task, and WMS digits-direct task. Higher leptin was also related to higher levels of phobic anxiety and worse performance on the IGT, WCST conceptual portion, and TMT. Finally, higher adiponectin values were correlated with lower scores on the UPPS-P lack of premeditation scale, and worse performance on the WMS direct and total scales.

### 3.4. Predictive Model for GD Presence

Table 3 shows the results of the logistic regression. The likelihood of being identified as GD was higher for individuals with lower education levels, lower social position indexes, greater psychopathological distress, higher impulsivity, lower self-transcendence, worse neuropsychological performance (specifically for WCST perseverative errors and on the Stroop color and WAIS vocabulary tasks), and lower LEAP2 levels.

## 4. Discussion

The present work studied gut hormones and adipocytokines, based on their association with reward and impulsive–compulsive processes, in people with GD compared with HCs. Likewise, neuropsychological, and clinical features were also evaluated, as well as its relationship with endocrine factors. Individuals with GD presented altered endocrine profiles compared to HCs, and regardless of BMI, were characterized by higher plasma ghrelin and lower LEAP2 and adiponectin concentrations, without significant differences in leptin levels. A worse neuropsychological performance, higher emotion dysregulation, greater psychopathological scores, higher impulsivity, and a more dysfunctional personality profile were also described in individuals with GD. Although significant correlations between endocrine factors and neuropsychological and clinical features were largely lacking, some neuropsychological domains and lower LEAP2 concentrations predicted GD presence. Implications are described below.

Increased plasma ghrelin concentrations in patients with GD seem consistent with results in SUDs, where ghrelin upregulation has been described [17,18], These findings suggest not only that this hormone could be involved in addictive processes [76] but also shared neurobiological substrates [3,4]. Despite not predicting GD presence, ghrelin up-regulation could speculatively contribute to maintenance of gambling due to its reinforcing properties [28], as well as being a risk factor for relapse, related to intensify craving [28]. If such possibilities received empirical support, similar to in SUDs, they may have important therapeutic implications for GD [27].

Although this is the first study to explore LEAP2 in GD, lower concentrations among individuals with GD suggest a possible dysfunction in the ghrelin system also involving LEAP2. The findings raise the intriguing possibility as to whether altered ghrelin production may influence LEAP2 release, supporting LEAP2 antagonism [22] and favoring lower LEAP2 concentrations. Moreover, both ghrelin and LEAP2 concentrations are subject to BMI in both animals and humans but in opposite ways [52]. Thus, persistent differences in ghrelin and LEAP2 after adjustment by BMI between groups, together with the finding that lower LEAP2 concentrations statistically predicted the presence of GD, suggest that these potential disturbances may be intrinsically associated with GD. Going one step further, our results raise the question of whether LEAP2 could be a potential therapeutic target in GD and other addictive-related disorders because of the neutralization of ghrelin’s possibly deleterious actions in craving, abstinence, and relapse. However, as LEAP2 has only been recently described and there is lack of extensive or consistent data in the literature, future research is needed [22].

The results regarding adiponectin agree with those reported in other addictive disorders [14,37]. Some protective functions have been linked to adiponectin, such as anti-inflammatory, anti-diabetic, and anti-atherogenic properties [77]. Thus, the results may in part explain a neurobiological basis for a worse metabolic state and a higher cardiometabolic risk associated with addiction, including individuals with GD, who had a significant higher BMI than HCs in our sample [77]. Speculatively, they may also in part explain incident cardiovascular conditions in relation to GD symptomatology in older adults [78]. Interestingly, our findings support the previous work by Geisel et al. [35], regarding leptin concentrations in GD. On the one hand, intrinsic compensatory mechanisms exist associated with endocrine dysfunctions in addiction, based on changes in receptors’ activity and/or hormones’ biosynthesis [79], which may be a possible rationale to explain the lack of differences between individuals with GD and HCs. Nevertheless, due to the heterogeneous methodology and mixed conclusions described in other addictive disorders, as well as limited work in GD, further studies are lacking to replicate these extend the current results.

Regarding neuropsychological performance, patients with GD had poorer cognitive flexibility and more perseverative errors than HCs, in line with previous findings [80]. Although we failed to find significant differences in the IGT trials, the GD group showed numerically less learning on the task, which may suggest a potential worse decision-making performance [41]. However, given the absence of statistically significant differences, the findings also resonate with prior reports showing similar patterns but no group differences in independent samples [81].

Patients with GD presented poorer estimated cognitive reserves compared to HCs. Lower scores on intellectual performance scales may be associated with a greater tendency to make risky decisions and may thus be a potential risk factor for the development of GD [82]. As a distinguishing finding, worse performance on the WAIS Vocabulary and WCST Perseverative errors predicted the presence of GD. Taken together, the results are in line with previous research suggesting that compulsive responding is in part mediated by impulsive decisions [39], since perseverative behavior has been “normalized” when feedback-response pause is increased in cognitive flexibility tasks [83]. One possibility is that low cognitive reserve may promote impulsivity, leading to an increase in perseverative behaviors in patients with GD. Even though significant correlations between endocrine and neuropsychological factors were largely absent, it may be worth further investigating possible common links based on relationships with reward-related neurocircuitry [84].

Patients with GD scored higher on general psychopathology and impulsivity measures [85], with more dysfunctional personality features (i.e., higher novelty-seeking and harm avoidance and lower reward dependence, self-directedness, cooperativeness, and self-transcendence) [85]. This profile has been linked to younger age of GD onset and problem-gambling severity [86,87]. Particularly, in our study, lower self-transcendence, and younger age, described as a possible risk factor of GD [2] also predicted the presence of GD. Self-transcendence seems to be a protective factor delaying the age of GD onset [86,88]. On the other hand, younger age has been positively linked to earlier GD onset, male sex, and higher novelty-seeking, and therefore, with problem-gambling severity [89]. Socio-demographic differences related to educational and socio-economic levels aligned with previous studies of our group [85]. From a social perspective, having a lower educational status and less social support have been previously implicated in GD [90].

Newly, relationships between endocrine and clinical variables were largely not observed. However, some previous studies revealed a relationship of appetite-related hormones with impulsivity domains and mood regulation [29,91]. Considering the complexity of addictive disorders and the limitations of cross-sectional studies, prospective studies with larger samples are warranted to understand better relationships over time.

### Limitations and Strengths

Some limitations should be mentioned. As the cross-sectional nature of this study limits causal attributions, future longitudinal studies are needed to better understand the involvement of neuroendocrine alterations and their roles in GD. Moreover, endocrine measurements were analyzed from peripheral blood samples, which could limit the inference of their functioning at a neural level. The lower number of individuals in the HC group with respect to the GD group may also limit the interpretation of the results, studies with a larger sample size are necessary to confirm the findings. Moreover, the GD group was principally composed of treatment-seeking males referred to a specialized unit in Catalonia, Spain. As such, studies of other compositions from other jurisdictions are warranted to determine generalizability of the results. Nonetheless, the representation of women in the study is consistent with the prevalence estimates in clinical treatment-seeking samples in GD, and comparable to their frequency in the control group. On the other hand, other strengths of this work is an adequate sample size, the well-characterized clinical and neuropsychological profile of both groups, and the adjustment in models for potentially confounding factors.

## 5. Conclusions

The present study provides evidence about underlying neuropsychological and endocrine dysfunctions related to reward processing in GD. The results have identified specific endocrine, neuropsychological, and clinical factors statistically predicting the presence of GD. Despite the cross-sectional design, this study supports a multifactorial nature of GD. Additionally, it supports the existence of potential neurobiological targets, known for involvement in other addictive disorders, with possible therapeutic implications. Hence, future research in this area may contribute to the development of more specific psychological and biological treatment strategies in GD.

## Figures and Tables

**Figure 1 nutrients-14-05084-f001:**
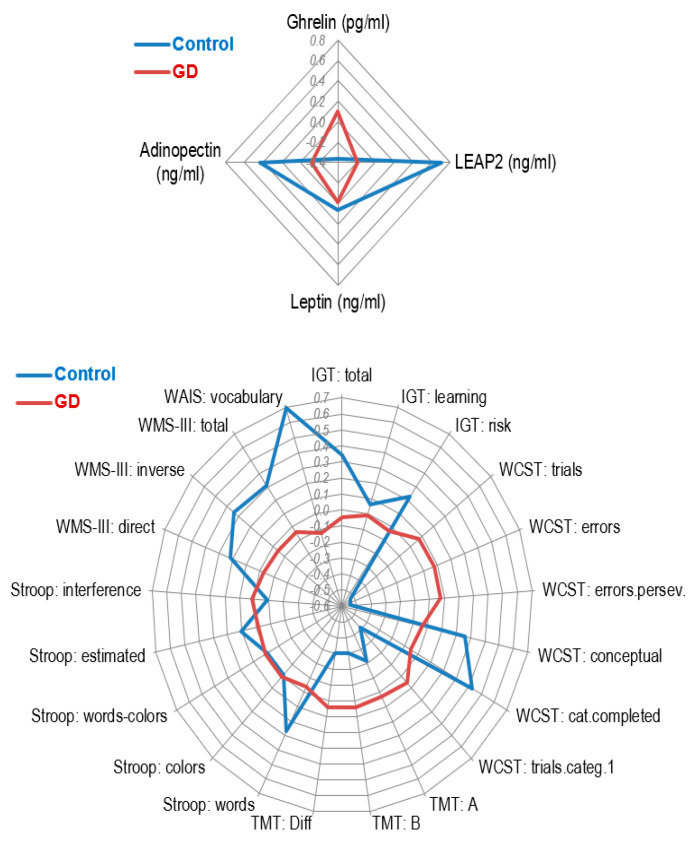
Radar-charts (z-standardized means are plotted). Note. GD: gambling disorder (*n* = 297). Control (*n* = 41). LEAP2: liver enriched antimicrobial peptide 2. IGT: Iowa Gambling Test. WCST: Wisconsin Card Sorting Test. TMT: Trail Making Test. WMS-III: Wechsler Memory Scale Third Edition. WAIS: Wechsler Adult Intelligence Scale. Due the different measurement scale for the variables in the graph, Z-standardized means are plotted to facilitate interpretation.

**Figure 2 nutrients-14-05084-f002:**
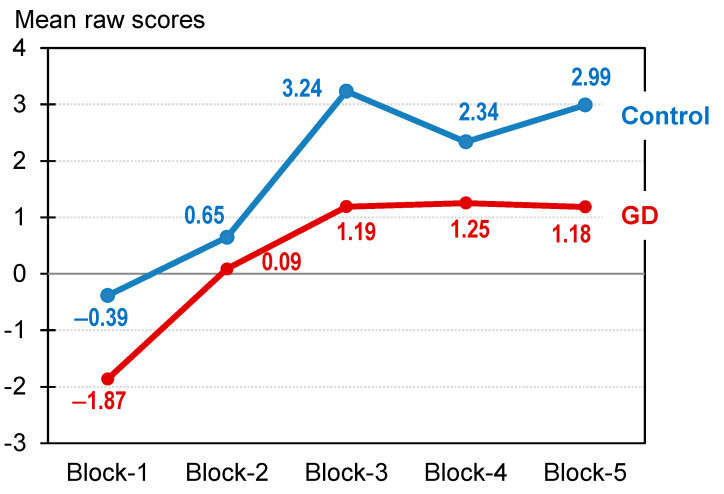
Performance-learning curve in the Iowa Gambling Test task. Note. GD: gambling disorder (*n* = 297). Control (*n* = 41).

**Table 1 nutrients-14-05084-t001:** Comparison of clinical characteristics via ANCOVA.

	Control (*N* = 79)		GD (*N* = 297)			
	Mean	SD	Mean	SD	*p*	|d|
1 Ghrelin (pg/mL)	544.92	673.59	958.48	753.26	**<0.001 ***	**0.58 †**
1 LEAP2 (ng/mL)	8.41	3.99	5.28	2.88	**<0.001 ***	**0.90 †**
1 Leptin (ng/mL)	9.00	8.13	8.18	7.85	0.402	0.10
1 Adiponectin (ng/mL)	12784.98	14084.20	8381.47	4374.29	**<0.001 ***	0.42
2 BMI (kg/m^2^)	24.99	2.36	26.57	5.04	**0.005 ***	0.40

Note. GD: gambling disorder. LEAP2: liver enriched antimicrobial peptide 2. BMI: body mass index. SD: standard deviation. |d|: Cohen’s-d coefficient. * Bold: significant comparison. 1 Adjustment by sex, age, and BMI. 2 Adjustment by sex and age. † Bold: effect size into the range mild-moderate (|d| > 0.50 and <0.80) to high-large (|d| > 0.80).

**Table 2 nutrients-14-05084-t002:** Comparison of the clinical characteristics via ANCOVA.

	Control (*N* = 41)		GD (*N* = 297)			
1 Neuropsychological Measures	Mean	SD	Mean	SD	*p*	|d|
IGT: block-1	−0.39	4.87	−1.87	5.16	0.115	0.30
IGT: block-2	0.65	5.87	0.09	5.52	0.583	0.10
IGT: block-3	3.24	9.50	1.19	6.97	0.126	0.25
IGT: block-4	2.34	9.85	1.25	7.46	0.446	0.12
IGT: block-5	2.99	8.33	1.18	8.56	0.249	0.21
IGT: total	8.88	28.10	2.10	21.94	0.104	0.27
IGT: learning	5.07	15.12	4.22	13.77	0.739	0.06
IGT: risk	5.33	15.78	2.44	13.64	0.257	0.20
WCST: trials	91.61	19.76	102.98	19.58	**0.001 ***	**0.58 †**
WCST: errors	20.04	16.28	33.14	21.77	**<0.001 ***	**0.68 †**
WCST: errors perseverative	9.13	6.27	15.09	10.06	**<0.001 ***	**0.71 †**
WCST: conceptual	65.77	8.66	60.60	16.25	0.063	0.40
WCST: categories completed	5.59	1.07	4.76	1.80	**0.006 ***	**0.56 †**
WCST: trials to complete 1-cat	17.64	6.52	26.79	28.09	0.053	0.45
TMT: A	28.81	8.10	31.77	10.51	0.088	0.32
TMT: B	70.67	22.07	78.41	36.38	0.202	0.26
TMT: Diff	41.74	18.48	47.75	32.35	0.271	0.23
Stroop: words	101.50	13.27	98.13	13.91	0.173	0.25
Stroop: colors	67.82	9.97	68.28	11.00	0.812	0.04
Stroop: words-colors	43.07	10.25	42.97	10.55	0.955	0.01
Stroop: estimated	40.48	5.20	40.11	5.53	0.700	0.07
Stroop: interference	2.59	7.71	2.86	7.75	0.838	0.04
WMS-III: direct	8.91	1.93	8.96	2.02	0.902	0.02
WMS-III: direct-span	5.99	1.12	6.01	1.16	0.947	0.01
WMS-III: inverse	6.55	1.89	6.18	1.99	0.304	0.19
WMS-III: inverse-span	4.80	0.98	4.64	1.15	0.427	0.15
WMS-III: total	15.46	3.36	15.14	3.62	0.617	0.09
WAIS: vocabulary	45.30	5.29	38.50	8.52	**<0.001 ***	**0.96 †**
2 Psychological measures	Mean	SD	Mean	SD	*p*	|d|
SCL-90R Somatization	0.43	0.35	0.99	0.78	**<0.001 ***	**0.92 †**
SCL-90R Obsessive/compul.	0.68	0.52	1.19	0.84	**<0.001 ***	**0.74 †**
SCL-90R Interp.sensitivity	0.40	0.38	0.99	0.80	**<0.001 ***	**0.95 †**
SCL-90R Depressive	0.51	0.59	1.54	0.93	**<0.001 ***	**1.32 †**
SCL-90R Anxiety	0.36	0.34	1.00	0.80	**<0.001 ***	**1.05 †**
SCL-90R Hostility	0.43	0.50	0.96	0.87	**<0.001 ***	**0.76 †**
SCL-90R Phobic anxiety	0.06	0.16	0.41	0.61	**<0.001 ***	**0.77 †**
SCL-90R Paranoid Ideation	0.46	0.47	0.95	0.79	**<0.001 ***	**0.75 †**
SCL-90R Psychotic	0.22	0.26	0.90	0.75	**<0.001 ***	**1.20 †**
SCL-90R GSI score	0.43	0.34	1.08	0.70	**<0.001 ***	**1.18 †**
SCL-90R PST score	26.37	16.45	47.53	20.77	**<0.001 ***	**1.13 †**
SCL-90R PSDI score	1.41	0.33	1.86	0.59	**<0.001 ***	**0.95 †**
UPPS-P Lack premeditation	20.98	4.02	24.32	5.51	**<0.001 ***	**0.69 †**
UPPS-P Lack perseverance	19.26	4.13	21.97	4.83	**0.001**	**0.60 †**
UPPS-P Sensation seeking	28.13	7.41	28.51	7.89	0.770	0.05
UPPS-P Positive urgency	20.70	5.85	31.92	9.22	**<0.001 ***	**1.45 †**
UPPS-P Negative urgency	23.02	5.55	32.25	6.44	**<0.001 ***	**1.54 †**
UPPS-P Total	112.13	18.36	138.83	22.37	**<0.001 ***	**1.30 †**
TCI-R Novelty seeking	99.34	10.63	110.82	13.13	**<0.001 ***	**0.96 †**
TCI-R Harm avoidance	88.04	17.86	98.79	16.83	**<0.001 ***	**0.62 †**
TCI-R Reward dependence	103.95	13.99	97.97	13.50	**0.009 ***	0.43
TCI-R Persistence	112.65	18.18	109.02	18.90	0.259	0.20
TCI-R Self-directedness	148.17	19.03	130.13	20.52	**<0.001 ***	**0.91 †**
TCI-R Cooperativeness	136.98	15.25	130.18	15.42	**0.010 ***	0.44
TCI-R Self-transcendence	66.73	15.93	61.38	13.83	**0.025 ***	0.36

Note. GD: gambling disorder. SD: standard deviation. IGT: Iowa Gambling Test. WCST: Wisconsin Card Sorting Test. TMT: Trail Making Test. WMS-III: Wechsler Memory Scale Third Edition. WAIS: Wechsler Adult Intelligence Scale. SCL-90R: Symptom Checklist-90-Revised. UPPS-P: Impulsive Behavior Scale. TCI-R: Temperament and Character Inventory-Revised. * Bold: significant comparison. 1 Adjustment by sex, age, and education. 2 Adjustment by sex and age. † Bold: effect size into the range mild-moderate (|d| > 0.50) to high-large (|d| > 0.80).

**Table 3 nutrients-14-05084-t003:** Predictive logistic regression model for identifying GD.

Dependent Variable: 1 = GD vs. 0 = HC	B	SE	*p*	OR	95% CI OR	
Covariates Sex (0 = women; 1 = men)	−0.781	1.498	0.602	0.458	0.024	8.623
Age (years-old)	−0.100	0.033	0.002	0.905	0.848	0.965
BMI (kg/m2)	0.508	0.149	0.001	1.662	1.241	2.226
Education (low levels)	2.875	0.754	0.001	17.724	4.045	77.665
Socioeconomic status (low levels)	1.099	0.543	0.043	3.000	1.035	8.696
Psychopathology distress (SCL-90R GSI)	2.483	0.896	0.006	11.973	2.069	69.290
Impulsivity (UPPS-P total)	0.082	0.023	0.001	1.086	1.038	1.135
Personality: TCI-R self-transcendence	−0.071	0.031	0.023	0.932	0.877	0.990
WCST Perseverative errors	0.180	0.068	0.008	1.198	1.048	1.368
Stroop Color	0.093	0.047	0.046	1.098	1.002	1.203
WAIS Vocabulary	−0.175	0.064	0.007	0.840	0.740	0.953
LEAP2 (ng/mL)	−0.326	0.126	0.009	0.722	0.564	0.923
Fit statistics	H-L = 0.985; R2 = 0.427; AUC = 0.986 (95% CI: 0.973 to 0.998)					

Note. GD: gambling disorder (*n* = 297). HC: healthy control (*n* = 41). Stepwise logistic regression adjusted by sex, age, and BMI. SE: standard error. OR: odds ratio. H-L: Hosmer–Lemeshow test (p-value). R2: Cox-Snell R2. AUC: area under the ROC curve (95% confidence interval (CI)). List of statistical predictors: sociodemographics (marital status, studies levels, and socioeconomic position), psychopathology distress (SCL-90R GSI), impulsivity level (UPPS-total), personality features (TCI-R), psycho-neurological profile, and endocrine measures (ghrelin, leptin, LEAP2 and adinopectin).

## Data Availability

Individuals may inquire with Jiménez-Murcia regarding the availability of the data as there are ongoing studies using the data. To avoid overlapping research efforts, Jiménez-Murcia will consider requests on a case-by-case basis.

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
