# Peer review of "Are Signals Regulating Energy Homeostasis Related to Neuropsychological and Clinical Features of Gambling Disorder? A Case–Control Study"

_nutrients, 2022, doi:10.3390/nu14235084_

Round 1

Reviewer 1 Report

The manuscript entitle” Are Signals Regulating Energy Homeostasis Related to Neuropsychological and Clinical Features of Gambling Disorder? Case-Control Study” investigated the neuropsychological and endocrine features in GD and showed the distinctive neuropsychological and endocrine dysfunctions might operate in individuals with GD and predicted GD presence. The manuscript is well-written; however, I have some comments that can improve the manuscript.

- In the material and method section, the inclusion and exclusion criteria should be more specific! In addition, other demographic data and their comparison between the two groups should be presented, particularly those that can affect the results, including education, marital status, BMI, and… 

- the time of blood sampling should be defined!! All samples collected at the same time?

- Although, I am not a professional in the field of Radar graphing! But it seems the data in table 1 and their illustration in figure I are different! For example, the concentration of Ghrelin in GD is less than twice of the Control group, but the figure shows a big difference!

- Abbreviations should be defined in the figure legend.

Reviewer 2 Report

The manuscript presents interesting data that can contribute to the scientific debate. However, the number of HCs participants represents a strong point of weakness.

 Abstract

The number of participants (experimental and control groups) is unbalanced and not even clear when reading the abstract. Unfortunately, even in the appropriate paragraph the authors failed to clearly define this fundamental methodological issue. Please make the number of participants clear.

Introduction

The reader would benefit from reading all diagnostic criteria for the GD.

Later on, the second aim needs to be improved, as it is is not precise.

Moreover, the rationale for neuropsychological tests and their choice needs to be explained and justified. In other words, which cognitive subcomponent, which cognitive ability the authors believe is related to the endocrine factors under consideration. What literature suggests those components and the tests identified by the authors. In the introduction, the authors refer to this literature (e.g., Granero et al) but in an overly concise and poorly delineated manner.

Participants

As abovementioned, the recruitment of the HCs is unclear and casts doubt on the methodological rationale.

The authors state they analyzed the performance considering sex/gender. How? As suggested by APA style, sex and gender should be not confused. Please, clarify this point.

Figures and Tables captions. All acronyms have to be given in the corresponding captions.

Discussion and limitations.

The authors must temper the discussion of their results in light of the low number of HCs. This limitation certainly undermined expectations. I suggest the authors discuss this point and/or propose analyses between two equalized subgroups
